# Deepint.net: A Rapid Deployment Platform for Smart Territories

**DOI:** 10.3390/s21010236

**Published:** 2021-01-01

**Authors:** Juan M. Corchado, Pablo Chamoso, Guillermo Hernández, Agustín San Roman Gutierrez, Alberto Rivas Camacho, Alfonso González-Briones, Francisco Pinto-Santos, Enrique Goyenechea, David Garcia-Retuerta, María Alonso-Miguel, Beatriz Bellido Hernandez, Diego Valdeolmillos Villaverde, Manuel Sanchez-Verdejo, Pablo Plaza-Martínez, Manuel López-Pérez, Sergio Manzano-García, Ricardo S. Alonso, Roberto Casado-Vara, Javier Prieto Tejedor, Fernando de la Prieta, Sara Rodríguez-González, Javier Parra-Domínguez, Mohd Saberi Mohamad, Saber Trabelsi, Enrique Díaz-Plaza, Jose Alberto Garcia-Coria, Tan Yigitcanlar, Paulo Novais, Sigeru Omatu

**Affiliations:** 1BISITE Research Group, University of Salamanca, 37007 Salamanca, Spain; chamoso@usal.es (P.C.); rivis@usal.es (A.R.C.); franpintosantos@usal.es (F.P.-S.); egoyene@usal.es (E.G.); dvid@usal.es (D.G.-R.); mamg@usal.es (M.A.-M.); beatriz_bellido@usal.es (B.B.H.); dval@usal.es (D.V.V.); Pablosk1997@usal.es (P.P.-M.); manulpb@usal.es (M.L.-P.); smanzano@usal.es (S.M.-G.); rober@usal.es (R.C.-V.); javierp@usal.es (J.P.T.); fer@usal.es (F.d.l.P.); srg@usal.es (S.R.-G.); javierparra@usal.es (J.P.-D.); 2Air Institute, IoT Digital Innovation Hub, 37188 Salamanca, Spain; guillehg@air-institute.org (G.H.); agustinsrg@air-institute.org (A.S.R.G.); mverdejo@air-institute.org (M.S.-V.); ralonso@air-institute.org (R.S.A.); 3Department of Electronics, Information and Communication, Faculty of Engineering, Osaka Institute of Technology, Osaka 535-8585, Japan; 4Research Group on Agent-Based, Social and Interdisciplinary Applications (GRASIA), Complutense University of Madrid, 28040 Madrid, Spain; alfonsogb@ucm.es; 5Institute For Artificial Intelligence & Big Data, Universiti Malaysia Kelantan, Kelantan 16100, Malaysia; saberi@umk.edu.my; 6Texas A&M University at Qatar, Doha 23874, Qatar; saber.trabelsi@qatar.tamu.edu; 7IBM, 28108 Madrid, Spain; enrique.diaz-plaza@es.ibm.com; 8Viewnext, 28036 Madrid, Spain; jalberto@usal.es; 9School of Built Environment, Queensland University of Technology, 2 George Street, Brisbane, QLD 4000, Australia; tan.yigitcanlar@qut.edu.au; 10ALGORITMI Centre/Department of Informatics, University of Minho, 4710-070 Braga, Portugal; pjon@di.uminho.pt; 11Digital Manufacturing Education and Research Center, Division of Data Driven Smart System, Hiroshima University, Hiroshima 739-8511, Japan; omatsu@hirosima-u.ac.jp

**Keywords:** smart cities, smart cyberphysical platform, data analysis, data visualization, edge computing, artificial intelligence, bike renting

## Abstract

This paper presents an efficient cyberphysical platform for the smart management of smart territories. It is efficient because it facilitates the implementation of data acquisition and data management methods, as well as data representation and dashboard configuration. The platform allows for the use of any type of data source, ranging from the measurements of a multi-functional IoT sensing devices to relational and non-relational databases. It is also smart because it incorporates a complete artificial intelligence suit for data analysis; it includes techniques for data classification, clustering, forecasting, optimization, visualization, etc. It is also compatible with the edge computing concept, allowing for the distribution of intelligence and the use of intelligent sensors. The concept of smart cities is evolving and adapting to new applications; the trend to create intelligent neighbourhoods, districts or territories is becoming increasingly popular, as opposed to the previous approach of managing an entire megacity. In this paper, the platform is presented, and its architecture and functionalities are described. Moreover, its operation has been validated in a case study where the bike renting service of Paris—Vélib’ Métropole has been managed. This platform could enable smart territories to develop adapted knowledge management systems, adapt them to new requirements and to use multiple types of data, and execute efficient computational and artificial intelligence algorithms. The platform optimizes the decisions taken by human experts through explainable artificial intelligence models that obtain data from IoT sensors, databases, the Internet, etc. The global intelligence of the platform could potentially coordinate its decision-making processes with intelligent nodes installed in the edge, which would use the most advanced data processing techniques.

## 1. Introduction

A smart city (SC) is an environment that uses innovative technologies to make networks and services more flexible, effective, and sustainable with the use of information, digital and telecommunication technologies, improving the city operations for the benefit of its citizens. However, where does the concept of SCs and smart territories stand currently? The United Nations Educational, Scientific and Cultural Organization (UNESCO) gives us a fairly accurate view on this situation: “All the cities and territories who claim the Smart City status are merely patchworks of opportunistic modernization, which is not always coherent and is sometimes juxtaposed without any real unity of function or meaning” [1]. This proposal has been fully developed on the basis of this statement and the PIs have accordingly guided the choice of the concepts to be developed in this project.

It is estimated that in 2030, the population density will increase by 30% in most cities, 60% of world population will live in cities, and there will be 43 megacities; these metropolitan zones will have more than 10 million residents [2]. Most experts agree on the fact that such population densities promote sustainable economic growth, which explains the increased mobility of the population from rural to urban areas. The downside of these advantages is the rise of uncontrollable sociological phenomena, such as urban violence and unhealthy crowding. Most cities face these issues, regardless of their political or economic regime. To counteract negative consequences, the emerging SCs have to adapt measures that will guarantee their economic attractiveness, and most importantly, they must meet the population’s high expectations regarding the quality of life.

Contrary to the initial smart city concept, which favoured the modernization of leading cities in developed countries, the new trend consists in deploying smart micro territories (or villages) within megacities and in their neighbouring regions, serving as smart satellites. The ideas presented in this proposal follow this SC deployment policy. The majority of SC projects aim to achieve sustainability and control economic growth by avoiding the loss of the already invested resources, by maintaining an ecology-friendly environment and by striving towards social equity. However, in reality this turns out to be very utopian, and several worldwide experiences show that this theoretical balance has never been successful at the practical level. As a result, SC development tends to be geared towards only one of all the objectives, failing to adapt a comprehensive approach.

We conceive the SC concept as a smart, realistic and technically balanced combination of the objectives and values described above. A SC should be thought out and deployed following the principles of modular design, where each module is implemented, tested and deployed independently so that it can be easily modified, replaced or exchanged. Thus, modules are defined as dynamic and evolutionary. Each module is dedicated to a particular task within the SC, such as Transportation Control, Logistics Planning, Traffic Control, Crowd Management, E-Health, etc. All modules interact and exchange the information collected within the SC through a centralized Management Platform. Deepint.net is the platform presented in this paper that is able to work independently and/or in collaboration with other (existing) SC platforms/IoT systems. Furthermore, the platform is equipped with the vast majority of the required connectors, facilitating its integration.

Such is the added value of the data that not only cities/territories but also users and companies in all sectors have been interested in AI-based data analysis and visualization methodologies. This is because they are fully aware of the benefits of those processes. As a result, the Artificial Intelligence sector has enjoyed high demand in recent years.

In this research, a platform has been developed that is capable of applying the most well-known techniques within the data analysis sector in a way that is simple and user-friendly. The platform’s design makes it fully prepared for the management of SCs and territories, regardless the size of the territories and the origin of the data. Deepint.net not only processes data, but also automates its intake, visualization and integration with any other platform and dashboard.

This platform is easy to use and does not require specialists in artificial intelligence, edge computing or machine learning. Deepint.net has been designed to provide managers with tools for data analysis and to help them generate models efficiently with no need for specialized data analysts or developers.

The platform aids the data analysis process at different levels: (i) it gives computer support so that cities do not have to invest in infrastructure; (ii) it offers mechanisms for the ingestion of data from different sources (relational and non-relational databases, files, repositories based on CKAN, streaming data, multi-functional IoT sensors, social networks, etc.) and in different formats; (iii) it offers data processing mechanisms to all users who do not have knowledge of programming (information fusion, data filtering, etc.); (iv) it offers information representation techniques based on interactive graphics to help understand data, the results of analysis and rapid decision making; (v) it helps select the methodologies that can be applied to the data provided by the user and automatically searches for the configuration that provides the best results (through cross-validation), so that the user does not have to configure anything at all. The data management flow can be found in Figure 1.

The platform, therefore, includes signal processing methods capable of transforming and using data (even real-time data) from any given source: IoT sensors, advanced multi-functional sensors, smart edge nodes, next generation networks, etc. or even the data from relational and non-relational databases. The platform helps the users select and use the most adequate combination of mathematical models; dynamic data assimilation and neurosymbolic artificial intelligence systems are capable of working with knowledge and data; they adapt to new time constraints and can explain why a certain decision has been taken. Research into the creation of explainable adaptive mixtures of expert systems is the key innovation of this project together with its application in the smart logistics field.

The platform has been used to develop several smart territory projects in the cities of Santa Marta and Carbajosa (Spain), in Caldas (Colombia), in Panama City, in Istanbul (Turkey), etc. The state of the art is reviewed in Section 2. Section 3 presents Deepint.net as a platform for managing smart territory projects. The results obtained from the use of the platform in Paris (France) are outlined in Section 4. Finally, the conclusions and future lines of research are described in Section 5.

## 2. Smart Territory Platforms and the Edge Computing Approach

It is expected that the world population will reach 9.7 billion in 2050. By then, two-thirds of the population will live in urban environments [1]. The United Nations estimates that by 2030 there will be 43 megacities (defined as metropolitan areas with a population greater than 10 million), most of them in developing countries. Critical social and ecological challenges that cities will face may include urban violence, inequality, discrimination, unemployment, poverty, unsustainable energy and water use, epidemics, pollution, environmental degradation, and increased risk of natural disasters.

The concept of SCs, which emerged in the early 2000s, attempts to provide solutions to these challenges by implementing information and communication technologies, improving the socio-ecological network of urban areas and the quality of life of its citizens. The initial concept of SCs focused on the modernization of megacities. However, most of the so-called SCs are just cities with several smart projects [1]. The main reason for this is that existing cities are difficult to modernize, mostly because buildings are too old to renovate, or they are heritage-listed buildings (due to their historical value), so they cannot be rebuilt or demolished. To overcome these limitations, different approaches have been proposed. The most promising trend is the creation of smart micro-territories, defined as hi-tech small towns, districts or satellite towns near megacities [1]. Real-life examples include the Songdo City; a satellite village near Seoul, or the Cyberabad District in Hyderabad (India). This paper present Deepint.net as a platform for the efficient and dynamic management of smart territories.

Given the availability of large amounts of data, the challenge is to identify intelligent and adaptive ways of combining the information to create valuable knowledge [3]. However, the implementation of SCs still poses several challenges, such as design and operational costs, sustainability, or information security. Thus, over the last few years, platforms have been designed and developed to provide innovative solutions to these problems. These platforms combine the data collected by electronic devices (sensors and actuators) with the data that has been generated by citizens and is stored on different types of databases.

In general, SC management platforms focus on one or several of the following dimensions: Crowd Management, Traffic Control and Smart Logistics, or resource prioritization in emergency scenarios, etc. These modules interact and exchange information among them. Figure 2 illustrates an architecture that is typical of these platforms, which normally work independently and/or in collaboration with other existing platforms and IoT systems.

Smart Mobility is probably considered one of the main dimensions of SCs. With the rapid population growth and its high concentration in urban environments, urban traffic congestion (which significantly lengthens waiting times) has a significant impact on the citizens’ daily life. To monitor and manage the state of road traffic, sensors are employed on road intersections and public transport vehicles, measuring location, speed, and density. Aimed at supporting local authorities in traffic control, recent studies propose different systems and platforms which improve the safety and security of the commuter [4,5,6,7,8]. Another important aspect of Smart Mobility is last-mile delivery. With the exponential growth of e-commerce, the logistics sector is experiencing efficiency difficulties. Technologies such as the Internet of Things or Autonomous Delivery Vehicles are expected to have a positive impact on this industry [9].

In a smart transportation system, crowd management is a key variable, not only during commutes and regular travel, but also when an event takes place. Major events with large crowd gatherings (sport events, concerts, protests, etc.) are celebrated in urban areas every year. Overcrowding and the poor management of crowds can lead to threatening and unsafe situations, such as injuries, stampedes and crushing. Therefore, effective crowd management is a crucial task. Crowd management systems have been developed to support services and infrastructures devoted to managing and controlling crowds at any time, so that in case of emergency situations, crowds are well managed, while the dangers and risks are minimized. Current studies focus on crowd counting and monitoring models [10] and algorithms [11], and crowd flow prediction architectures [12], their aim is to provide the government and local authorities with valuable information on large crowds [13].

The objective of many SC projects is to prioritize the efficient use of resources: irrigation systems, energy consumption, or dealing with emergency scenarios, such as the one provoked by the current pandemic. Thousands of people have lost their lives in epidemics, pandemics, natural disasters (hurricanes, floods, fires) and human-induced disasters (stampedes, terrorist attacks or communicable diseases), and it is believed that a reasonable number of these fatalities are associated with the poor management of crowds and slow response to accidents. One of the key challenges involved in managing an emergency is minimizing the time it takes for the personnel and supplies to arrive at the scenario. Emergency services, such as health services, police, and fire departments, are expected to make critical decisions and to correctly prioritize the use of resources, using limited time and information. To facilitate this task, several systems have been proposed in recent years. Rego et al. [14] introduced an IoT-based platform that modifies the routes of normal and emergency road traffic to reduce the time it takes for resources to arrive at the scenario of an emergency. Rajak and Kushwaha [15] propose a framework capable of creating a “Green Corridor” for emergency vehicles. Ranga and Sumi [16] present a traffic management system that helps ambulances and fire trucks find the shortest routes.

Social media data could be of much value when responding to an emergency. Alkhatib et al. [17] propose a novel framework for the management of incidents in SCs by using social media data; Perez and Zeadally [18] present a communication architecture for crowd management in disruptive emergency scenarios; and Kousiouris et al. [19] propose a tracking and monitoring system that identifies events of interest for Twitter users.

Statistics from the Department of Economic and Social Affairs of the United Nations, DESAP, indicate that 68% of the world population will live in cities or urban areas by 2050 [20], which means rapid and even uncontrolled growth with consequent challenges for governments, for example: pollution, limited mobility due to traffic and congestion; high cost of housing, food and basic services; as well as security problems [21].

To address this growth, the SC concept emerged as the integration of the urban environment with information and communication technologies (ICTs), attracting the interest of all major sectors (governments, universities, research centres, etc.) in presenting solutions or developments that would make up a SC [22]. The objective of this paradigm is the effective management of the challenges associated with the growth of urban areas through the adoption of ICTs in developments, solutions, applications, services, or even in the design of state policies [23].

Currently, the term SC is widely used, for example in the systematic reviews of the literature there are more than 36 definitions that address different dimensions of the urban environment such as: mobility, technology, public services, economy, environment, quality of life or governance [24]. One of the most widely used definitions is the one proposed by Elmaghraby and Losavio [25]: “an intelligent city is one that incorporates information and communication technologies to increase operational efficiency, shares information independently within the system and improves the overall effectiveness of services and the well-being of citizens”. However, the growth of the Internet of Things and of the devices permanently connected to the Internet has led to a growing interest in data management [23] and security [26] as new urban management challenges emerge.

Ensuring the security of information, devices, infrastructure and users in an environment where large volumes of data are managed in real time is the objective of state-of-the-art research, because it is a critical element of any solution aimed at smart cities/territories [20]. This challenge has generated a research trend: Edge Computing and its integration with IoT, which is reflected in statistics and in the interest of large corporations in research, development and implementation opportunities in smart territory scenarios, so that they can increase their profits and market shares [27,28,29,30,31,32,33]. Table 1 lists the researches that have employed Edge Computing in different SC scenarios. These proposals evidence the interest in this technology.

In this context, most cities are not prepared and do not have the policies required to understand and ensure the confidentiality of a huge amount of data, as well as their correct processing and storage. Another important factor is the application of artificial intelligence techniques for the extraction of information which facilitates the management of key infrastructures, systems and devices in a district, making them functional and efficient. However, rapid response time is fundamental for the functioning of a Smart Territory. In addition, the large volume of data that is sent from a Smart Territory directly to the cloud has high associated and variable costs, forcing cities to seek solutions that reduce the cost of using cloud services, energy and bandwidth.

In a Smart Territory scenario, the proposed architecture should be capable of managing the heterogeneity of the IoT devices in order to ensure the management system’s safety and efficiency. Although there are many proposals in the state of the art, Figure 3 represents a general model of most architectures, where Blockchain technology may be implemented to preserve the safety and reliability of the sensitive data generated by IoT devices at the edge of the network. Edge nodes are also included in architectures of this type, using artificial intelligence and deep learning algorithms for filtering, real-time data processing and low latency [34].

Figure 3 shows a basic schema of an edge computing architecture with edge nodes that allow user application processes to be executed closer to the data sources [35]. The edge nodes perform computing tasks such as filtering, processing, caching, load balancing, requesting services and information, and reducing the amount of data that are sent or received from the cloud.

### Smart City Vertical Markets and Tools

The development of a Smart City involves a series of phases, from planning to the selection of the most suitable tools. Both the citizens and the private sector have to be involved in the development of a Smart City, resulting in a connected, innovative, digital and successful city.

The projects that constitute a Smart City can be classified as follows: Smart Governance, Smart Economy, Smart Mobility, Smart Environment, Smart People and Smart Living. On the basis of these indicators, a number of rankings classify the most advanced Smart Cities, which are the fruit of governance oriented towards innovation and digital inclusion. The highest scores have been obtained by London, Singapore, Seoul, New York and Helsinki. It should be noted that these cities also stand out for having one of the best open data portals, another Smart City pillar that favours innovation. In particular, the high quality of data available in several US cities has led to a number of Data Science competitions on the Kaggle platform to predict future demand for city bikes.

In addition, Europe’s profits from the use of AI-based software are expected to reach a value of more than $1.5 billion by 2025, five times the amount obtained in 2020. Several companies have presented proposals related to smart city platforms and architectures, highlighting the Huawei Horizon@City, Toyota and NTT smart city platform and IBM City Operations Platform reference architecture proposals. The Deepint.net tool has been created as the starting point for the solution to these problems, facilitating the rapid and efficient development of the infrastructure that any Smart City requires. Smart City vertical markets and their domains can be found in Table 2.

## 3. Deepint.net Platform for Smart Territories

Today, cities/territories are the largest data producers, and all major sectors can extract knowledge and benefit considerably from data analyses. Thanks to advances in computing such as distributed processing techniques, improved processing capabilities and cheaper technology, current artificial intelligence techniques can be applied to large volumes of data at a very fast pace; this would have been unthinkable less than a decade ago.

As a result, large investments are being made in the information and computing sector, either through the acquisition of technology that allows for the recovery and processing of information, or by investing in hiring highly qualified scientists to carry out precise studies. Such staff is not always easy to find due to the scientific complexity involved and the peculiarities of the problem domain.

Deepint.net is a platform that seeks to cover the current "gap" between the need to create smart territories and the big expenses that this normally entails in terms of tools and data scientists. This platform has been created for the managers of intelligent cities/territories, facilitating all aspects of data management, processing and visualization.

Deepint.net is a platform deployed in a self-adapting cloud environment, which enables users to apply artificial intelligence methodologies to their data using the most widespread techniques (random forest, neural networks, etc.), even if they lack knowledge of their operation/configuration, or even programming skills. Deepint.net facilitates the construction of models for data processing, guiding the user through the process. It indicates how to ingest data, work with the data, visualize the information, apply a model and, finally, obtain, evaluate, interpret and use the results. The platform incorporates a wizard that automates the process, it is even able to select the configuration for the artificial intelligence methodology that will provide the best solution to the problem the user is trying to solve.

Furthermore, additional features enable users to exploit all the results through dynamic, reusable dashboards that can be shared and used by other SC tools. Moreover, the results can be exported in different formats for simple integration, for example, in a report. Figure 4 shows elements and tools that can be used along the process of data management, from data intake, to the creation of scorecards or data exploitation.

Deepint.net is a platform created for managing and interpreting data in an efficient and simple way. It has been structured in five different functional layers as shown in Figure 4. Figure 5 presents the elements of the data ingestion layer of Deepint.net.

On Deepint.net, both static and dynamic data may be incorporated in the tool. Dynamic data are constantly updated. The data are stored as ’data sources’ as presented in Figure 5. The data sources on Deepint.net are elementary because they are the starting point of the rest of the functions that can be applied. To create a data source from data that are available elsewhere, the wizard asks the user to specify the type of media in which the original data is found, and the configuration associated with that type of media. For example, if it is a database, the user must indicate the host, username, password, database name and the SQL query to be executed. After indicating the configuration of the data source, the wizard requests information on the data update frequency in case of dynamic data. Finally, it is possible to encrypt the data in a data source on Deepint.net. This option slightly slows down all operations, as the user is asked to perform a decryption operation every time they want to make use of the data. Nevertheless, it provides an extra layer of security that other tools on the market do not offer.

Regarding the type of support in which the original data are found, the following are allowed: (i) direct sources: CSV or JSON files containing the data to be imported from local files, URLs or calls to existing endpoints; (ii) derived sources: new data sources obtained from existing data sources (very useful for the next step of the flow, data management); (iii) databases: both relational and NoSQL databases; (iv) Other services: for data coming from well-known services such as AWS S3, CKAN or data streaming (such as MQTT).

Deepint.net offers multiple functionalities to users for the management of the information contained in data sources (Figure 6). To begin with, the system automatically detects the type of data (Appendix A) and the format (for decimal data or dates), thus, the user does not have to spend time on specifying it. However, in the case of certain graphs or models it may be important to specify the type of data and Deepint.net allows users to specify it manually or change the type that has been detected automatically. It also allows the user to generate features from existing fields using user-defined expressions. At this point in the flow, the creation of derived data sources from existing data sources is possible, as discussed in the previous section. More specifically, different types of operations may be performed on the different data sources, such as filters on records or parameters in a data source, merging two data sources with the same parameters, and much more. The tool also offers the possibility of working with data sources from an API to edit them programmatically.

The platform provides tools for knowledge extraction within the context of various learning methodologies, as shown in Figure 7. We can find in it multiple supervised learning methodologies, both for classification and regression problems, using algorithms such as Decision Tree, Random Forest, Gradient Boosting, Extreme Gradient Boosting, Naive Bayes, Support Vector Machines, and linear and logistic regressions. In all of the cases, the configuration of the algorithms can be adjusted to achieve better performance. Additionally, there are unsupervised learning techniques available, including clustering methods (k-means, DBSCAN, and others), as well as association rule learning or dimensionality reduction techniques, for example PCA.

Another field of application of the platform is Natural Language Processing, which involves processes such as text classification, text clustering, and similarity-based retrieval.

The tool offers a wizard that facilitates the process of creating dynamic and interactive graphs. It only takes a few simple steps, as presented in Figure 8. In the first step, the user is asked to specify the data source they want to use to create the visualization. They can create as many visualizations of a data source as they wish. In the same step, the source can be filtered to represent a subset that meets the conditions specified by the user (conditions can be nested with AND and OR operations). Similarly, the user can select a subset of the sample, which is ordered randomly or by the user in situations where large volumes of data are represented with pivot charts which may slow down the user’s computer (since these are pivot charts developed in JavaScript, the processing power is provided by the client). The next step that the user has to do is to select the type of chart they want and configure it. The configuration depends specifically on each of the types of graphs and there are more than 30 different graphs. Likewise, the information to be represented can be configured, the style (title, legend, series, colors, etc.) can be easily set using the wizard.

Once the visualizations are created, they can be added to interactive dashboards and placed in the positions that the user wants by means of drag and drop. Different forms can be added with which the user can interact to filter the information that is represented (a second level of filtering in addition to the filtering performed when creating the visualization). This offers multiple benefits when it comes to controlling cities and monitoring only the information that is relevant at any given time. The user can also add other elements to the dashboards, such as the results of the machine learning models, iframes, images or content through WYSIWYG editors, among many other possibilities.

A very important aspect of data analysis is being able to export the obtained results, as well as the data used as input. This feature facilitates, for example, the use of other types of tools and enables the scientific community to reproduce the system. Deepint.net allows to export all data sources to CSV or JSON files, as well as the results of the developed artificial intelligence models or visualizations (such as static PNG images) to, for example, be able to incorporate them in documents or reports (Figure 9).

However, one of the most powerful features of the tool is the possibility of sharing the dashboards that have been created, both with the users of the tool, and with those that receive a unique link from the user. Through this link, all the functionality incorporated in the dashboards can be accessed in real time and it is even possible to integrate the dashboards in third-party tools through the use of iframes or WebViews, for example.

Figure 10 shows some screenshots of data ingestion. In short, the platform covers all the usual flow of data analysis from the intake of information to the exploitation of the results. However, unlike other existing tools, its user does not need to have any knowledge of programming or data analysis. Figure 11 shows a screenshot of the process of creating a supervised model for data analysis.

Deepint.net offers mechanisms for the management of all the information provided by the user, enabling the creation of different projects. Figure 12 shows a screenshot of the process of creating a model for data visualization, which as you can see, is extremely simple, since it consists of selecting the type of model to display the data, the data set and the selected parameters. Figure 13 gives a screenshot of the dashboards created for the visualization of city information, Panama (top) and Istanbul (bottom). Similarly, the creation of users and permissions is allowed so that all the group members/employees can exploit the results of the analysis, displaying them on dynamic and interactive dashboards. Deepint.net is a versatile, multipurpose platform, whose utility for smart territory or city management is of special interest.

These are some of the functionalities of Deepint.net:User management functionality which offers plans that are customized to the needs of the cities.Integration of multi-source data, prioritizing the most common sources: formatted local and internet (CSV/JSON) files, databases (NoSQL and SQL), streaming data (MQTT, among others), CKAN-based repositories, etc.Automatic detection of the data type to facilitate analysis and representation.Mechanisms for data processing (filtering records according to a criterion, eliminating fields, merging sources, creating compound fields, etc.).Guided mechanisms that facilitate the representation of the information provided by the user.Mechanisms for the guided creation of data analysis models, suggesting the best configuration to the users while allowing advanced users to carry out this process themselves if they wish.Simple evaluation of the results of the model according to different metrics.Dashboard definition by inserting created visualizations, model results, etc. through ’drag and drop’ so that users can customize how they want to work with the tool.Structuring the user’s projects so that with one account the tool can be used in different areas or for different clients.Creation of city users with different roles so that all employees can use the platform as specified by the administrator.Exportation of results for easy integration in reports, etc.Possibility to deploy the system in a commercial cloud environment (AWS) that allows to provide services on demand to all users, in a way that is adapted to their needs, with high performance and high availability.

Compared to its competitors, one of the main advantages of this platform are wizards. Users just need to learn how to use them, which does not require any advanced knowledge, in order to integrate the results of machine learning tools, the monitoring tools and the visualization of results. The data accuracy of all data science tasks depends directly on the input and the selected algorithms. The accuracy of both generic and specific algorithms is considered constant independently of the platform. The volume of data that the platform can manage depends on its architecture, which is introduced in the section that follows.

### Platform Architecture

The platform architecture can be deployed in an on-premise environment or in a commercial cloud environment.

The on-premise solution has been designed for situations in which it is not possible to process information in the infrastructures of third-party companies due to, for example, restrictive data protection policies.

Nevertheless, the solutions advertized on the platform web page are all hosted in commercial cloud environments (other solutions require a custom study and deployment).

Figure 14 provides a high-level representation of the architecture to be deployed in AWS (Amazon Web Services). Clients can connect to the application through the internet, available on app.deepint.net. The load balancer redirects the traffic to the corresponding EC2 instance. The users of the free version share resources, while the users of the paid version have private EC2 instances and do not share resources, which guarantees a good processing capacity at all times.

In each EC2 there is a web server and task workers, which are managed by a Redis server for event management on the platform (for the cache and the pub/sub system).

These resources access the serverless systems when they stop dealing with information. For the processing of information on Deepint, a relational database is used, in this case an Aurora DB as it is the serverless relational system of AWS. In addition, the system uses the storage system S3 to deploy all the data sources which the users upload on the system. The deployment can be encrypted with AES-256 if the user specifies it.

As the cloud environment of AWS is used, there is no limit on the volume of data and the response times depend directly on the type of EC2 chosen by the client. In case the client needs to improve the response times, they can increase their expenditure on resources or migrate to an on-premise solution.

The main technical challenges considered during the platform design are: parallelization aspects, availability of on-demand resources and a serverless solution which allows the user to work on with no size restrictions.

## 4. Case Study

This platform can be adapted for the development of vertical markets, as part of SC management. In general, vertical markets are associated with mobility, security, pollution, etc.

The following is a case study on the use of the platform to develop a system for bicycle rental management in Paris. It allows the user to identify the areas of Paris in which they are more likely to find bicycles using historical data in order to predict areas with the highest bicycle density in real-time. The process is carried out using the Pareto optimal location algorithm.

### 4.1. Pareto Optimal Location Algorithm

In this section, we present an algorithm for optimal geographically-distributed resource selection.

Let ${\left\{C_{i}\right\}}_{i}$ be a set of locations offering the resources a user is seeking. For example, the resources might be public bike sharing stands. The user is interested in borrowing a bike, but they risk arriving at a location where there are no bikes available. The probability of one of those resources can be modeled as a set of functions ${\left\{p_{i}\left(t\right)\right\}}_{i}$, where t is the time.

Such functions can be approximated using an existing dataset, where further dependence on other variables is allowed. For example, a set of models ${\left\{p_{i}\left(weekday,hour,weather\right)\right\}}_{i}$ can be built using machine learning prediction algorithms. An example made with the Vélib dataset, available from the open data Paris portal, is shown in Figure 15.

While these models can be used to suggest the resource locations to the user, to maximize the probability of satisfying their need, it is necessary to address the interplay between distance to the resource and the likelihood of finding an available bike. A notion of Pareto optimality [36] can be introduced to reduce the choice to a smaller set of points. A resource location *j* dominates another one *i* if the following conditions apply:(1)$$\left\{\begin{array}{c} (d\left(x,C_{i}\right)\geq d\left(x,C_{j}\right)\\ p_{i}\left(t+d\left(x,C_{i}\right)\right)\leq p_{j}\left(t+d\left(x,C_{j}\right)\right) \end{array}\right\}\Leftrightarrow C_{i}\leq C_{j}$$
where *d* is the estimated time to arrive at a given location, *x* is the user’s location, and, in the right-hand-side, a notation for this domination condition has been introduced. A location $C_{i}$ in a set $\left\{C_{i}\right\}\left(i\in I\right)$, Pareto is optimal if there is no other point dominating it, i.e.,
(2)$$∄j\in I\;:\;j\neq i,C_{i}\leq C_{j}$$

A natural approximation for simplifying the model assumes the variations of probability are negligible on the time scale of displacement, i.e.,
(3)$$p_{i}\left(t+d\left(x,C_{i}\right)\right)\approx p_{i}\left(t\right).$$

This allows to reduce the notion to a regular Pareto condition on the plane $\left(d\left(x,C_{i}\right),p_{i}\right)$. An example is shown in Figure 16, where optimal points are found in the upper left corner.

A representation of the points in terms of real-world coordinates is shown in Figure 17.

Finally, an ordered list of recommendations can be provided to the user by introducing a risk-aversion parameter ω∈R, so
(4)$$v_{i}=p_{i}-\frac{d(x,C_{i})}{\omega }$$
defines a metric which serves to order the Pareto optimal points. Users with higher risk-aversion (larger ω) tend to choose options with greater probabilities, while users with higher risk-tolerance (smaller ω) tend to choose the closest resources. The choice of reasonable values for ω is scale-dependent and can be tuned by presenting sets of Pareto optimal points to the user and asking for their preference.

### 4.2. Implementation with Deepint.net

The algorithm described in the previous section was implemented using the Deepint.net platform, as well as specific deployments. An outline of the process is described in this section.

Firstly, a prediction algorithm was built using tabulated historical data. A simple example of such data is shown in Figure 18. These data could be used to predict the probability of availability (ratio) as a function of the station, the weather conditions, and the date (here only distinguishing weekdays and weekends for simplicity).

Regressors were built using these data with the assistance of the platform’s online wizard. Figure 19 shows a Random Forest model, including a predicted-observed diagram and an interactive form to invoke the model.

Finally, a specially designed mobile application used the Deepint.net API (Figure 20) to retrieve the model predictions. This information was used to perform the Pareto optimization as described in the previous section, providing the user with an interactive map where they could choose their preferred option.

The integration of the platform was completed with the construction of a set of dashboards which allowed to monitor the information, as shown in Figure 21.

## 5. Platform Evaluation, Conclusions and Future Work

Deepint.net is a platform that facilitates knowledge management and the creation of intelligent systems for managing territories efficiently. The platform facilitates the use of centralized intelligence and edge architectures, with intelligent nodes, allowing for both decentralized and centralized analyses.

The implementation of smart territory management systems involves a reduction in costs associated with maintenance and resource management. Depending on its application, the platform can facilitate traffic optimization, create systems for analyzing the opinions citizens on social networks, or help assess and prevent pollution, etc.

In general, the use of a platform of this type, which allows for the use of any cloud, reduces initial investment needs. Normally, costly infrastructures are required to analyze medium and large volumes, since the cost can be very high if 1GB is exceeded. This cost would be significantly reduced by processing the data in a remote cloud-based infrastructure tailored to the needs of each territory. The use of commercial infrastructures also reduces risks and increases the security of data management. Moreover, it is possible to scale the infrastructure to the needs of each moment. The platform is designed to ingest and manage any type of infrastructure, such as intelligent nodes, facilitating the decentralization of intelligence and the creation of intelligent models distributed in edge computing mode.

Similarly, the territories or cities in which this platform would be used would not need to have staff with programming or data analysis knowledge, Instead, they would only need to have knowledge of the information owned by their company. The user takes on the role of a data analysis expert; they work with the data and understand if the obtained results are satisfactory or not. This allows to focus on the result and not on the development costs, which would be negligible thanks to the proposed system.

The system may be operated in real time by any user in the city. Users with no computer knowledge would only display the information on a real-time dashboard while managers would be in charge of monitoring the general performance of the company. All city stakeholders can benefit.

The wizards offered by Deepint.net for the integration of data sources, creation of visualizations, dashboards and modelling, cover the entire ecosystem within the data analysis life cycle. This proposal has advantage over other commercial data analysis solutions which are much more limited in functionality and usability.

The development process of this platform has been made efficient thanks to the use of Deepint.net. The platform also facilitated the carrying out of the case study in the city of Paris. Moreover, Deepint.net is high versatile and it is currently undergoing further development in order to incorporate new functionalities which can adapt to a wider range of smart territories, eventually aiming to become a comprehensive agent capable of speeding up the development of any smart city. Furthermore, in-depth research is being carried out on the many technologies which form deepint.net and the findings will be shared with the scientific community in future articles.

## Figures and Tables

**Figure 1 sensors-21-00236-f001:**
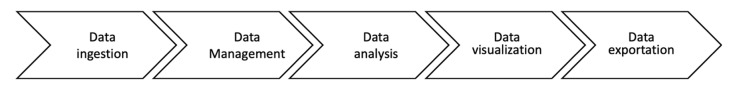
Data management flow in Deepint.net.

**Figure 2 sensors-21-00236-f002:**
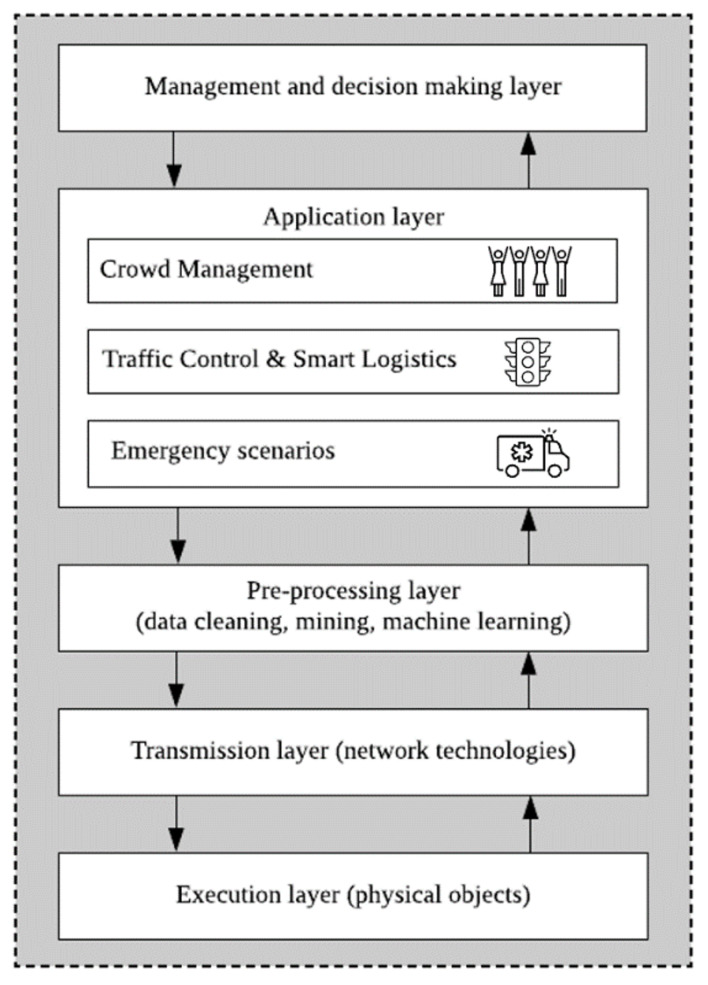
Smart city (SC) management architecture.

**Figure 3 sensors-21-00236-f003:**
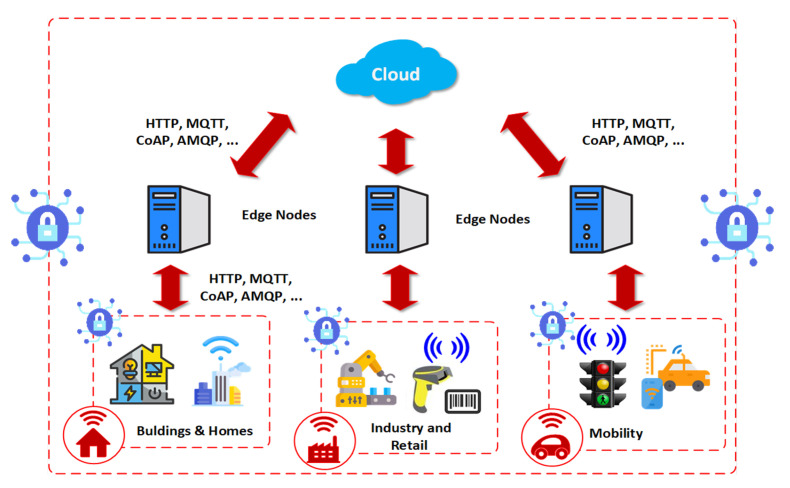
Edge Computing for SCs [34].

**Figure 4 sensors-21-00236-f004:**
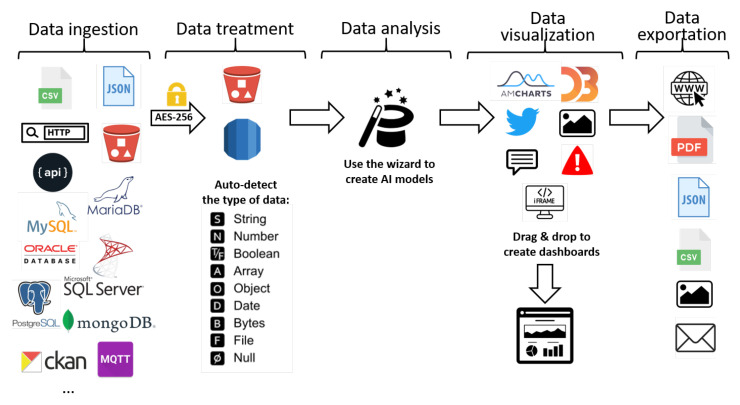
Data analysis flow and elements.

**Figure 5 sensors-21-00236-f005:**
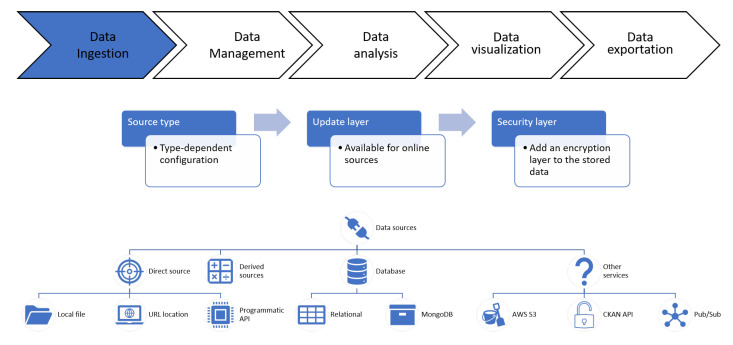
Data ingestion layer description.

**Figure 6 sensors-21-00236-f006:**
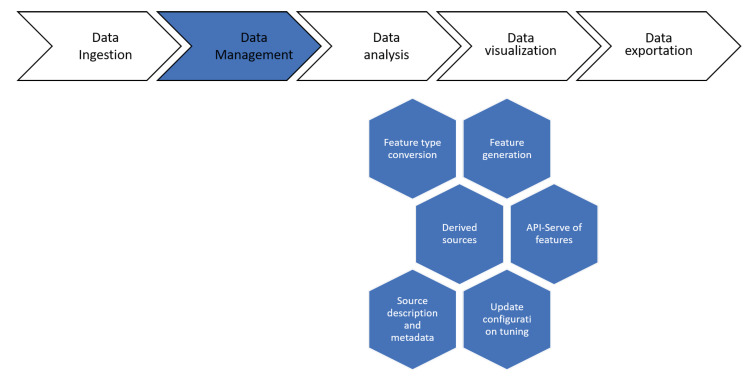
Data management layer.

**Figure 7 sensors-21-00236-f007:**
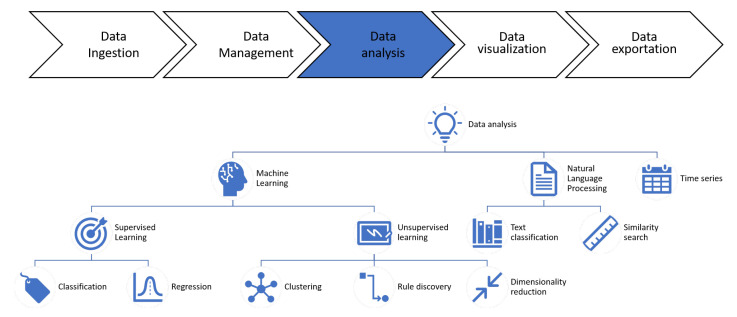
Data analysis layer.

**Figure 8 sensors-21-00236-f008:**
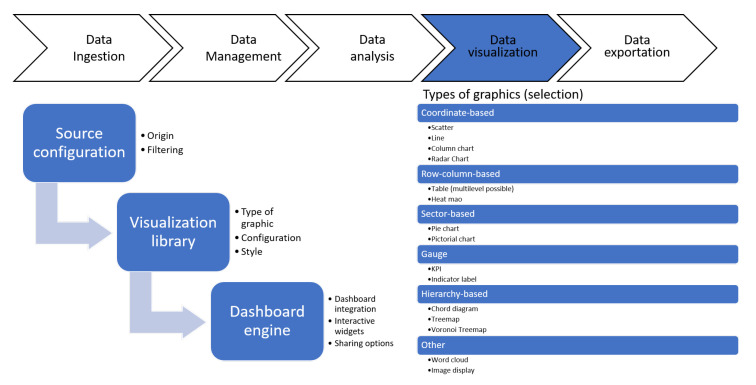
Data visualization layer.

**Figure 9 sensors-21-00236-f009:**
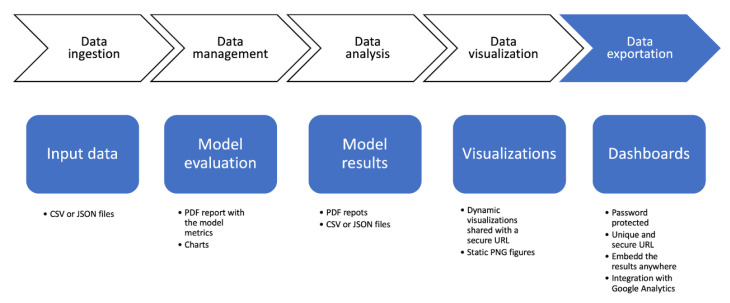
Data exportation layer.

**Figure 10 sensors-21-00236-f010:**
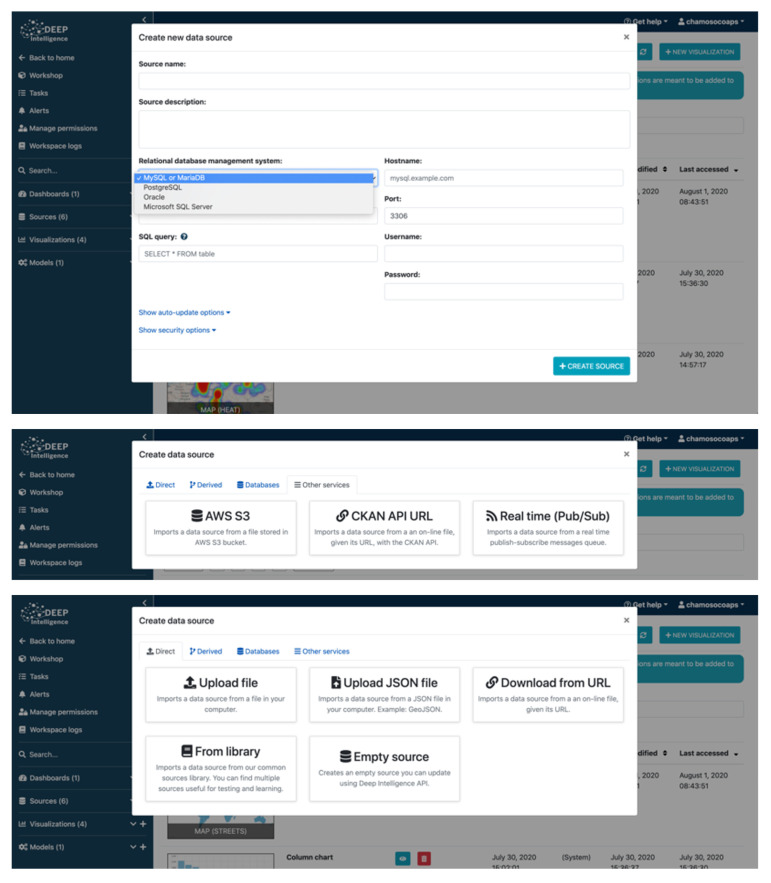
Data ingestion.

**Figure 11 sensors-21-00236-f011:**
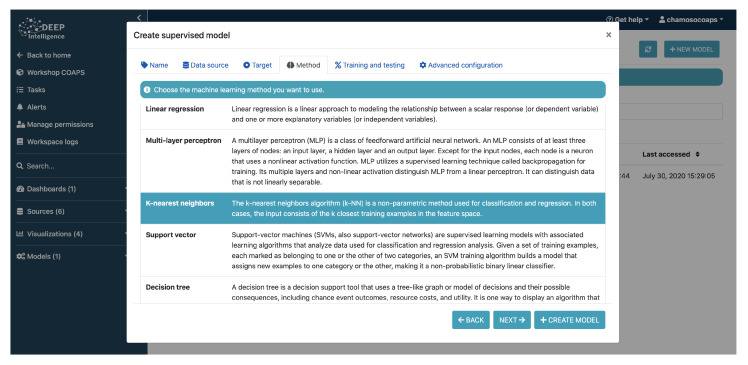
Intelligent model selection.

**Figure 12 sensors-21-00236-f012:**
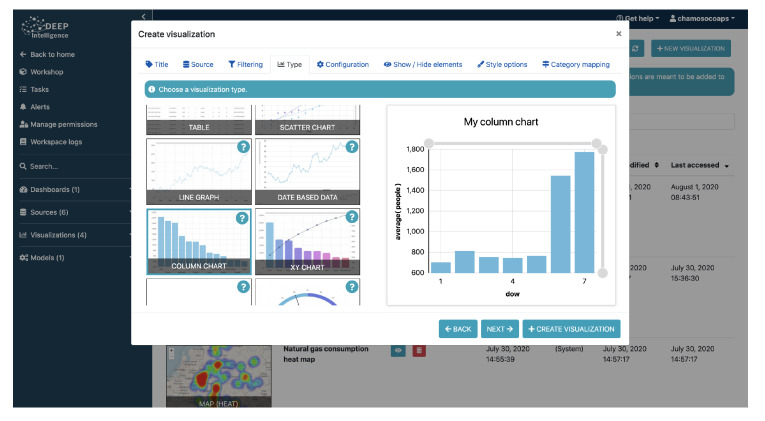
Screenshot of the process of creating visualization models.

**Figure 13 sensors-21-00236-f013:**
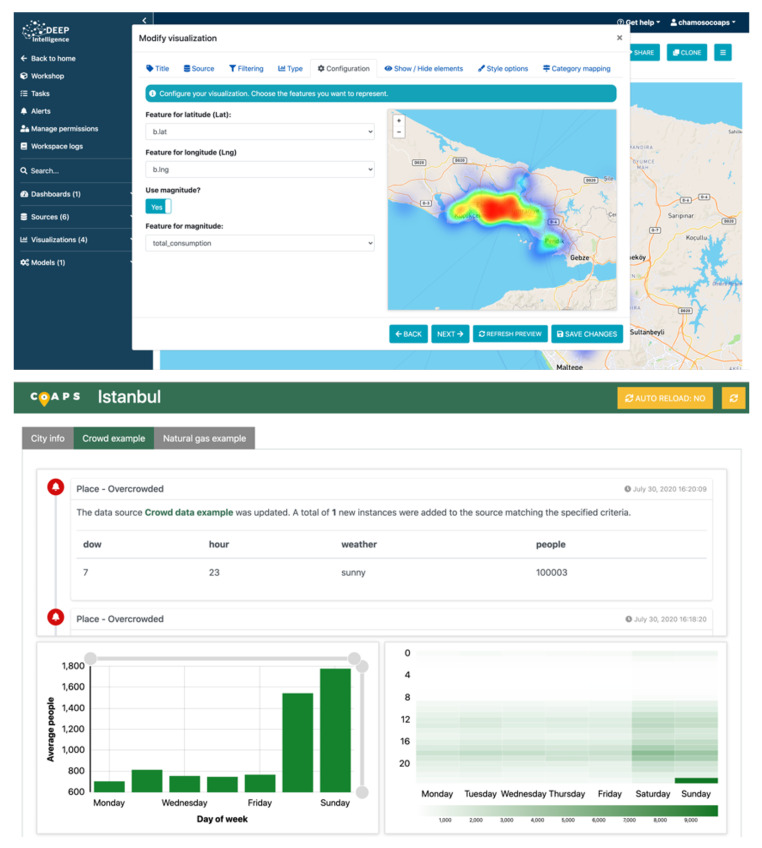
Creation of dashboards. Panama (**top**) and Istanbul (**bottom**).

**Figure 14 sensors-21-00236-f014:**
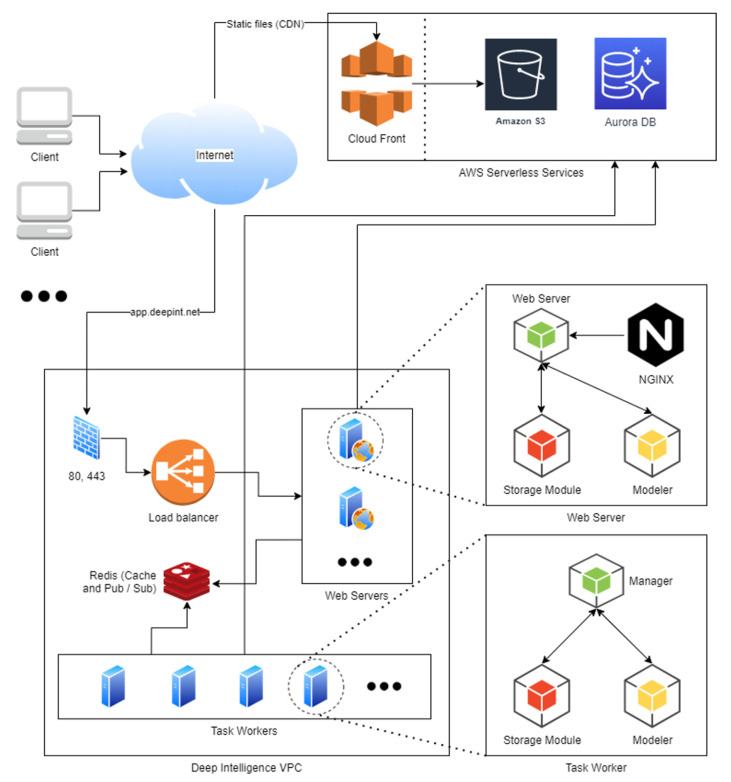
High-level representation of the platform architecture.

**Figure 15 sensors-21-00236-f015:**
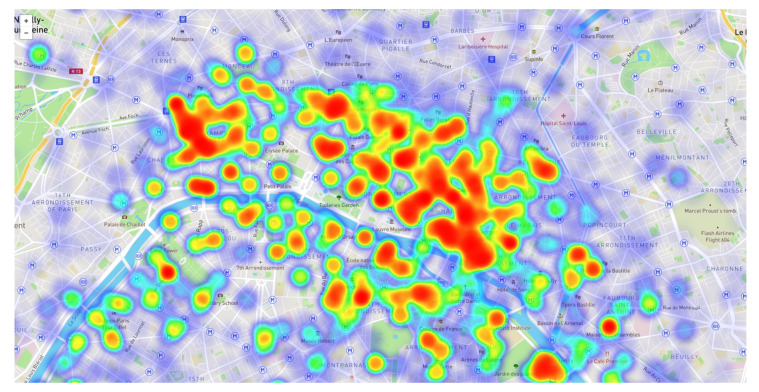
The heat map of a probability model for the availability of bikes in the city of Paris. A smoothed plot has been generated by overlapping Gaussian kernels weighted with the estimated availability probability.

**Figure 16 sensors-21-00236-f016:**
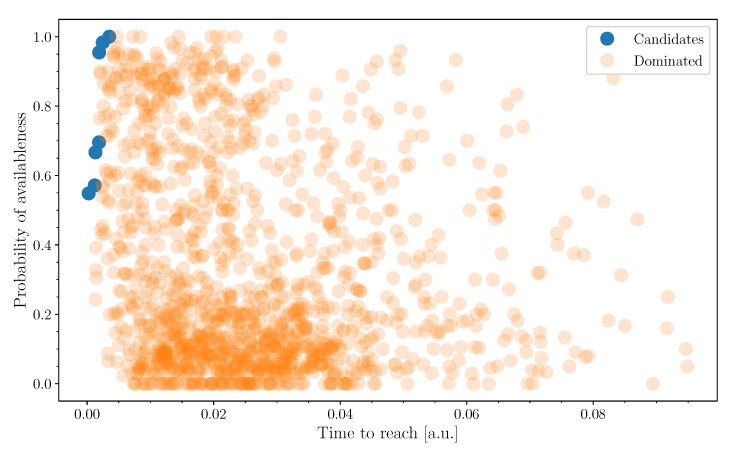
Pareto optimal points on the time-probability plane.

**Figure 17 sensors-21-00236-f017:**
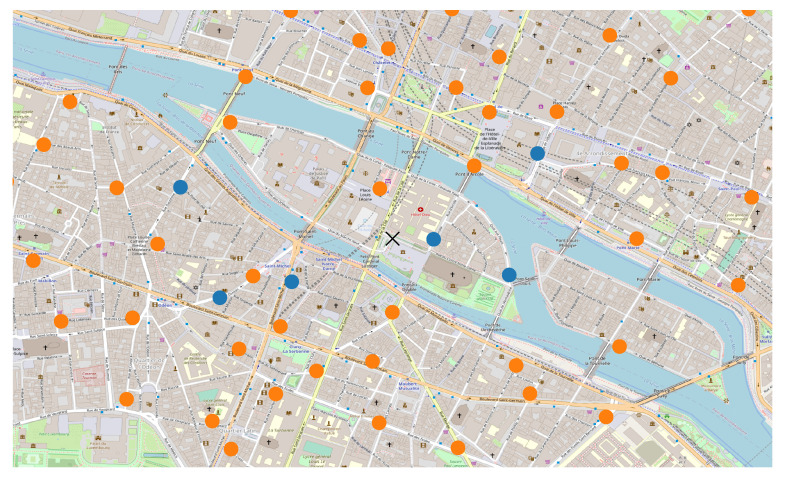
Real-world coordinates of the Pareto optimal points. The location of the user is shown as the black cross. Pareto-optimal choices are shown as blue dots, while sub-optimal locations are shown in orange.

**Figure 18 sensors-21-00236-f018:**
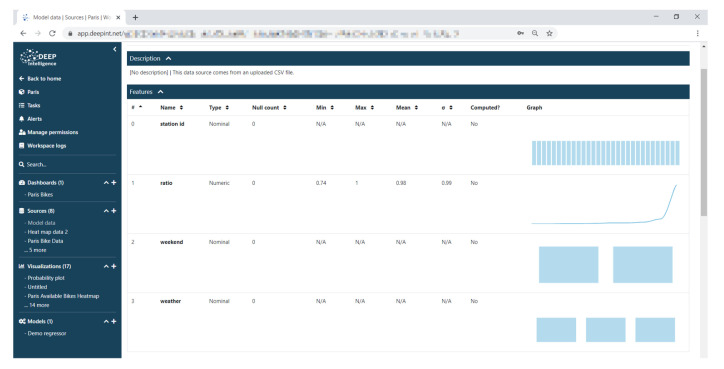
Data uploaded on Deepint.net to build a prediction model.

**Figure 19 sensors-21-00236-f019:**
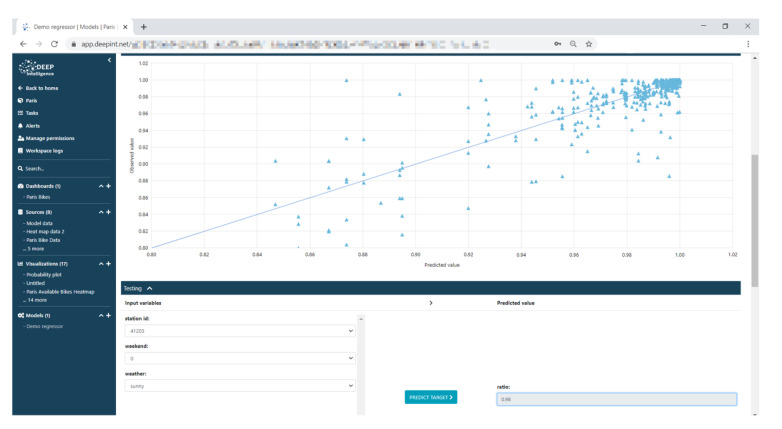
Prediction model built on Deepint.net.

**Figure 20 sensors-21-00236-f020:**
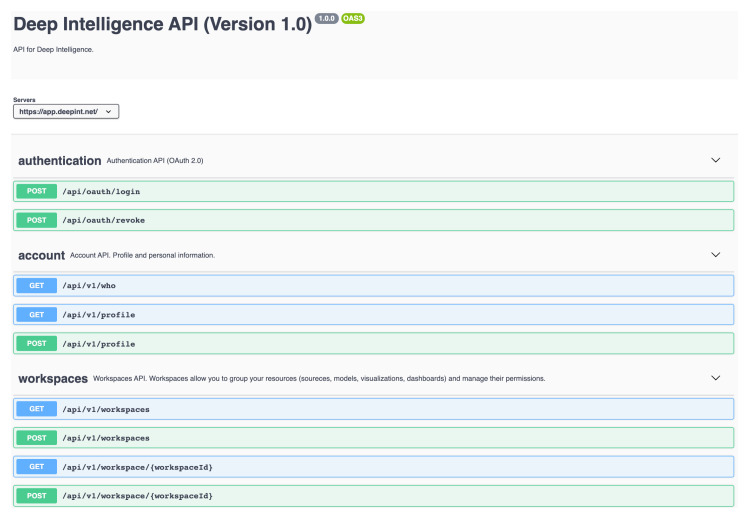
Online documentation of the Deepint.net API.

**Figure 21 sensors-21-00236-f021:**
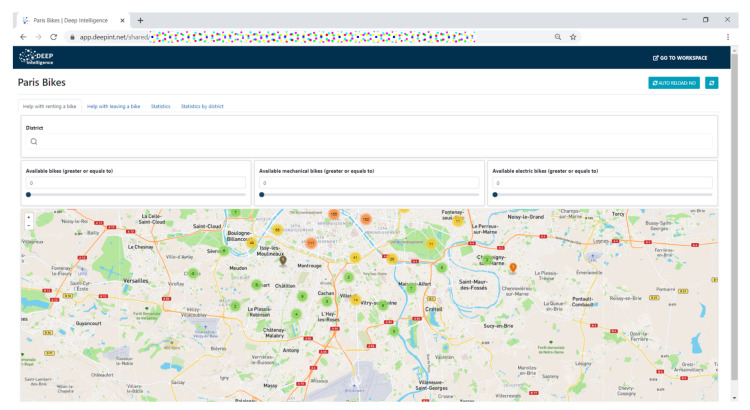
System dashboard built using the Deepint.net platform.

**Table 1 sensors-21-00236-t001:** Edge Computing applied in SC/Territory environments.

Field	Solution	Author
**Mobility**	· The study considered a SC/Territory scenario where vehicles ran applications that take data from the environment and send them to edge computing servers through roadside units.	Premsankar et al., (2018)
	· The authors conduct a case study on the fog and edge computing requirements of the intelligent traffic light management system.	Hussain et al., (2019)
**Tourism**	Mobile edge computing potential in making cities/territories smarter.	Taleb et al., (2017)
**Industry and Augmented Reality**	Proposes the use of Edge Computing to enable intelligent management of industrial tasks.	Schenider et al., (2017)
**Smart District (manhole cover)**	Edge computing servers interact with corresponding management personnel through mobile devices based on the collected information. A demo application of the proposed IMCS in the Xiasha District of Hangzhou, China, showed its high efficiency.	Jia et al., (2018)

**Table 2 sensors-21-00236-t002:** The vertical markets and domains of Smart Cities.

Vertical Markets	Domain
**Smart Governanca**	Participation Social Services Transparency
**Smart Economy**	Innovation Productivity Entrepreneurship Flexible labour market
**Smart Mobility**	Connected public transport Multimodality Logistics Accessibility
**Smart Environment**	Environmental protection Resources management Energy efficiency
**Smart People**	Digital education Creativity Inclusive society
**Smart Living**	Tourism Security Healthcare Culture

## Data Availability

Data available in a publicly accessible repository. The data presented in this study is openly available in app.deepint.net.

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
