# Peer review of "Deepint.net: A Rapid Deployment Platform for Smart Territories"

_sensors, 2021, doi:10.3390/s21010236_

Round 1

Reviewer 1 Report

The paper introduces a novel online platform for development and tracing of IoT projects at a large scale, that is, at the level of smart environments in neighborhoods and cities. Smart cities constitute an actual and promising field for research and development, and instruments like the proposed one can be very helpful in the management of upcoming projects.

The platform implements some of the most known machine learning tools, as well as visualization tools that can be applied for the presentation of the obtained results.

The paper is devoted to the exposition of challenging efforts on smart territories, and to the presentation of the online platform. Then, a case study of the implementation of two projects is illustrated as an example of the platform advantages.

It is not really a scientific paper, since there is no novel development in the document, but an informative brochure about the goodness, functionalities and possibilities of the platform.

In the Introduction section, it is stated that in 2030, there will be 43 megacities. This assertion could be flexibilized, since there is no full evidence of this number to be exact.

Since the list of authors is so extensive, a final section describing the individual contribution of each author should be interesting.

There are some minor typos throughout the text. The platform name, Deepint.net, is misspelled in some points of the text:

Page 3, line 72: "Deep.int"

Page 7, line 235: "deepint.net" (no capital letter). Same occurs in page 19, line 447.

Page 2, line 26: "the city's operations" --> "the city operations"

Page 3, line 89: "time constrains" --> "time constraints"

Page 8, table 2: It seems to be a missing "markets" after "Vertical".

Page 8, line 263: "Figure 4 shows that elements and tools..."; there is an extra "that".

Page 15, line 376: "the platform's use" --> "the platform use"

Author Response

Dear Editors and Reviewers,

We are very grateful for the time spent on carefully handling and reviewing our submitted paper. Based on your valuable comments, we have significantly updated the previous version of this paper to make it better.

The reviewers’ comments are listed along with our responses and modifications in the attachment.

Thank you very much again for your kind help.

The authors.

Reviewer 2 Report

This paper describes an efficient network physical platform for the smart management of smart territories. It is not only efficient and intelligent but also compatible with edge computing concepts, allowing intelligent distribution and the use of intelligent sensors. This paper introduces the platform throughout and describes its architecture and functions. The results obtained from the use of the platform in Paris (France) are outlined in the fourth part of the article.

  1. At the beginning of the article, it is generally not allowed to have so many authors in the paper. Counting down, there are 29 participants in the article.
  2. Please show the data formats allowed by the platform in the appendices instead of simply explaining in the text that any data format is allowed.
  3. Throughout the article, the citation format of the article does not meet the requirements of sensors. One of the reasons is that the order of citing documents is not obvious.
  4. In the introduction part of the article, the author frequently uses words like "All." Is the expression too absolute? For example, all the cities and territories, in all cities, all experts, all SC projects, all the objectives, all modules, all the required connectors, and so on.
  5. Please highlight the advantages of the platform by comparing some of the latest platform technologies and features. Also, what is the theoretical basis of focus analysis?
  6. The platform's general description seems to occupy most of the article, but there is no in-depth introduction of the detailed platform technology and its operating principle.
  7. The abstract of the article mentions that the platform optimizes the decisions taken by human experts through explainable artificial intelligence models. So how is the interpretability of the model reflected in the article?
  8. The fonts in some pictures are too small, such as figure 5, which will inevitably cause some inconvenience in reading.
  9. When the platform aids the data analysis process, how does the platform automatically look for and provide the best result configuration?
  10. The article involves too many conceptual words, and it seems that the core content is rarely introduced, making it easy to understand.
  11. There are so many related technologies involved in this platform that the article's focus is not prominent. It is recommended that these technologies be described in sections or divided into several papers to describe the key points.
  12. What is the model used for each function of the platform, and what are the platform's advantages compared with other similar websites?

Author Response

(The authors gave the same response as above.)

Reviewer 3 Report

Authors developed the Deepint.net platform to assist SC to grow quickly. It is a nice website. This paper can be published if extra info can be added:

  1. Please separately evaluate your platform such as data accuracies in predictions/volume/response time..... and a comparison table from other platforms would be a big plus.
  2. Specify your platform architecture.
  3. A typo: section 4.4--->section 4

Author Response

(The authors gave the same response as above.)

Round 2

Reviewer 2 Report

The authors proposed ideas for an efficient cyber-physical platform for the smart management of smart territories. It is efficient because it facilitates data acquisition and data management methods and data representation and dashboard configuration. The platform allows for the use of any data source, ranging from the measurements of multi-functional IoT sensing devices to relational and non-relational databases. It is also smart because it incorporates a complete artificial intelligence suite for data analysis; it includes data classification, clustering, forecasting, optimization, visualization, etc. It is also compatible with the edge computing concept, allowing for the distribution of intelligence and the use of intelligent sensors. The concept of smart cities is evolving and adapting to new applications; the trend to create intelligent neighborhoods, districts or territories is becoming increasingly popular, as opposed to the previous approach of managing an entire megacity. The platform is presented, and its architecture and functionalities are described. Moreover, an example is given of its application in managing the bikes renting service of Paris - Vélib’ Métropole. This platform's development could allow smart territories to develop adapted knowledge management systems, adapt them to new requirements, use multiple types of data, and execute efficient computational and artificial intelligence algorithms. The platform optimizes the decisions taken by human experts through explainable artificial intelligence models that obtain data from IoT sensors, databases, the Internet, etc. The platform's global intelligence could potentially coordinate its decision-making processes with intelligent nodes installed in the edge that would use the most advanced data processing techniques. I recommend the following revisions. Besides, some questions need to be explained below:

  1. Avoid lumping references as in [x-z] and all others. Instead, summarize the main contribution of each referenced paper in a separate sentence. For scientific and research papers, it is unnecessary to give several references that say precisely the same. Anyway, that would be strange. Since then, what is the innovative scientific contribution of referenced papers? Each thesis state only one reference.
  2. English language should be carefully checked, and carefully check the paper for language typos.
  3. The use of English must be improved.
  4. Avoid using the first person.
  5. When the platform aids in analyzing data, the parameters are set by default to values, which researchers have found to work well in general. Please show through the experimental data results that the default value can achieve good results in other data sets. If not, I think this statement is not very convincing.
  6. The paper should emphasize the description and detailed solution process of difficult technical problems of designing the platform instead of describing the platform operation instructions and platform function introduction.
  7. In the manuscript, the Figures' quality is inferior; please replace the Figures with high-quality ones. (for example, Figures 2, 3, 5, 10, 11, 12, 13, 14, 15-21.)

Author Response

Dear Reviewer,

We are very grateful for the time spent on carefully handling and reviewing our submitted paper. Based on your valuable comments, we have significantly updated the previous version of this paper to make it better.

The reviewers’ comments are listed along with our responses and modifications in the attachment.

Thank you very much again for your kind help.

The authors.
